# Addressing Disability Inequities: Let’s Stop Admiring the Problem and Do Something about It

**DOI:** 10.3390/ijerph191911886

**Published:** 2022-09-20

**Authors:** James H. Rimmer

**Affiliations:** School of Health Professions, Lakeshore Foundation Research Collaborative, University of Alabama, Birmingham, AL 35206, USA; jrimmer@uab.edu

**Keywords:** wellness, health promotion, health disparities, health inequities, disability, disabling conditions

## Abstract

The healthcare system and public health community are often underprepared to support the needs of people with disabilities and to include them equitably in wellness programs (e.g., exercise, leisure, nutrition, stress management) offered to the general community. Consequently, the vast majority of people with disabilities are unable to make the transition from “patient” to “participant,” which contributes to many of the health disparities reported in this population. People with disabilities have a disproportionately higher rate of acquiring secondary conditions such as obesity, cardiovascular comorbidity, pain, fatigue, depression, deconditioning, and type 2 diabetes, often resulting from poor access to home and community-based health promotion/wellness programs that include physical activity, nutrition, stress reduction, and sleep hygiene, among others. Achieving health equity in people with disabilities requires a multi-stage approach that includes person-centered referral to wellness programs, empowering people with disabilities to become self-managers of their own health and ensuring that community-based programs and services are inclusive. A three-stage model for addressing health and wellness needs across the home and community settings is described, which is currently being used in a large federally funded center in the US with potential generalizability across the world.

## 1. Introduction

People with disabilities are an unrecognized disparity population [1] that experience health inequities rooted in a history of systemic discrimination and exclusion [2]. With reference to the U.S., we have yet to find solutions to the societal responsibilities and barriers that prevent individuals with disabilities from leading healthy, active lives. Simply put, we have not achieved health equity. Working-age adults with disabilities visit their healthcare provider and the emergency department more frequently than people without disability [3], and individuals with disabilities are seldom exposed to wellness programs that focus on improving their quality of life and helping them prevent or manage physical and psychosocial secondary health conditions [4]. The secondary conditions associated with disability, overlapping the natural course of aging, can bring an onset of new medical issues across an individual’s lifespan increasing the occurrence of health disparities to a much great degree compared to the general population.

While having a disability does not equate with being unhealthy, individuals with disabilities do experience a greater incidence of poor health than other underserved groups and people without disability [5]. In the U.S., more than 80 percent of people with disabilities experience one or more chronic health conditions [6,7,8,9]. Across their lifespan, they are at increased risk for cardiovascular comorbidity (e.g., obesity, impaired glucose tolerance, deconditioning), [10] chronic pain, [11,12] and anxiety, depression, and social isolation [13,14]. In the U.S., 38.9% of individuals with disabilities are obese, compared to only 26.1% of their non-disabled peers aged 18 and over [10]. Similarly, Similarly, 16.3% of people with disabilities are diagnosed with diabetes, in contrast to 7.2% of people without a disability [10].

The healthcare system in the U.S. has not been designed to support the transition of people with disabilities through the stages of care from initial injury or diagnosis to lifelong health and wellness. Typically, conventional healthcare systems in the U.S. have a narrow interpretation of health that focuses on managing a specific health condition or disease process versus promoting self-managed wellness [15,16]. Almost 25 years ago, Ryff and Singer noted that the focus of health in Western medicine is on “disease, illness and medical concepts” and not “rates of wellness and positive functioning” [17]. Not much has changed in the current climate of healthcare in the U.S. This presents major challenges to people with disabilities and/or their family members in transitioning from healthcare services into a holistic wellness program that supports their need for addressing mind, body, and emotional/spirit-based health dimensions. Moving beyond medical models of disease care into wellness care is greatly needed for most populations, but there is an even greater need among people with disabilities. They often experience multiple roadblocks in self-managing their health (e.g., lack of transportation to facilities and programs, unaware of existing programs, unable to pay for a program, etc.), [6,18,19] and these social, cultural, environmental and attitudinal barriers make it extremely difficult to lead an active and healthy lifestyle.

Understanding how to expand the continuum of healthcare services into the domains of wellness in order to provide people with disabilities more opportunities to self-manage their physical, mental and emotional/spiritual health was the rationale for this viewpoint. As suggested in the title, there is no shortage of papers and reports documenting health disparities/inequities in people with disabilities; however, there continues to be a pervasive lack of published wellness/health promotion programs tailored or adapted for people with disabilities and sustainable over the long term. While this viewpoint focuses on one approach in the U.S., largely due to one federal agency’s support to promote and sustain health/wellness in people with disabilities through a national center that has been funded for more than two decades, it is important for readers to understand that there are other countries (the number unknown) that have also focused on *integrative* wellness directives for people with disabilities including the Netherlands [20,21] and Australia [22,23]. To be clear, there are numerous countries that have targeted physical activity/exercise (one domain of physical wellness) for people with disabilities. However, the emphasis of this paper is to highlight the need for integrative, holistic wellness that, in addition to exercise, includes other areas of wellness encompassing the body, mind, and spirit. Moreover, there are likely several other countries who have unpublished wellness initiatives and programs for people with disabilities that have not been reported in the mainstreamed literature, and in studies that have been published, without long-term support for sustaining them, they are often not available on a national or international level. This paper should not be misconstrued as a global perspective on the needs of wellness in people with disabilities, but rather the start of a dialogue on how the U.S. has made a substantial commitment in federal resources to promote the interconnectivity between healthcare and wellness for Americans with disabilities. 

## 2. Disability and Wellness Inequities Start in Healthcare

The current cultural mindset in healthcare has been structured around a medical model focused on curing disability rather than living a fulfilling life with disability [24]. Patients (the term only used to describe people with disabilities accessing healthcare services) may focus solely on the hope of recovery, which has been perpetuated by the medical model, rather than on the potential for maintaining health and life satisfaction while living with a disability. People with a recently acquired disability or new diagnosis may perceive their condition as a limitation in self-managing their health. They may have few role models to emulate and lack access to an active group of peers. Many patients may become socially isolated leading to a deficit of self-managed wellness skills. Socially isolated individuals present higher risks of morbidity and mortality from all causes of disease/chronic conditions [14].

The gap between healthcare and wellness fosters disability inequities [4,25]. The inherent disconnect between the two professions—one working on the front end treating health issues (e.g., pain management, mental health) and the other (i.e., community health promotion providers) working on the back end preventing/managing chronic health conditions—may lie at the root of health inequities observed in people with disabilities [26]. Healthcare providers rarely direct patients with disabilities to wellness programs because they either do not have the time or are unaware of what programs or resources may be appropriate for their patients. Physicians who typically have high patient loads must focus their encounters on immediate medical services such as discussing new and existing health conditions (e.g., new type of pain, higher levels of fatigue, significant weight gain, more anxious), recommending additional diagnostics, and/or prescribing medications. Some primary care physicians may not feel comfortable addressing certain physical or cognitive conditions associated with the disability [27], which adds to the challenge of patients with disability moving beyond specialized medical care into self-managed wellness [8].

Information on wellness initiatives is also not adequately integrated into the healthcare system. Healthcare providers that do routinely advise patients about the health benefits of exercise and diet are likely not knowledgeable on offering similar types of advice for people with disabilities [26]. Most resources about wellness in doctor’s offices are not tailored to people with disabilities. The unintended consequence is that the vast majority of people with disabilities accessing healthcare are unable to make the transition from “patient” to “wellness participant” [4] and disability and wellness inequities end in the home and community.

Poor access to home and community-based health promotion/wellness programs likely has a major effect on the health disparities reported in different subgroups of people with disabilities [11,28]. The numerous barriers that they are exposed to in managing or improving their health have been documented in multiple studies across levels of socio-ecological model (SEM). A multifactorial set of barriers affect the individual (e.g., lack of tailored wellness recommendations to address certain physical limitations and secondary conditions such as pain and fatigue), interpersonal (e.g., lack of social support, relationships leading to social isolation), institutional/organizational (e.g., lack of inclusive wellness services offered by healthcare providers to people with disabilities), community (e.g., poorly designed outdoor spaces and street connectivity), and structures and systems (e.g., limited or no transportation services to exercise and recreation venues) [18]. Financial barriers also prevent many individuals from benefiting from wellness initiatives [19,29]. Even when individuals are connected to appropriate wellness programs, the cost to participate can be prohibitively high for some people on a fixed budget. Communities that have focused on policy, environmental, and programmatic changes to promote active living, healthy eating, and weight management seldom address access issues for people with disabilities [1,30].

## 3. Addressing Disability Inequities from Home to Community

The healthcare and public health systems must enable people with disabilities to transition from medically managed healthcare to full- and co-managed wellness (health coach-participant dyad), to self-managed wellness (participant only), with formal transitions between each of these levels as the essential linkages between a healthcare-to-home-to-community model. Addressing disability inequities in people with disabilities must start with a comprehensive set of wellness options offered at home or in the community, access to health coaches when needed to address certain questions, and broader access to, and inclusion in, existing evidence-based home and community-based wellness programs and public health initiatives that afford people with disabilities multiple options for improving their health.

Beginning with a home-based telewellness program eliminates certain barriers that impede the ability of people with disabilities to get to a community wellness program. Because many community barriers limit opportunities for people with disabilities to engage in community-based wellness programs, it is critical that they have options to engage in wellness in the comfort of their home (especially during the COVID-19 pandemic). Lack of transportation, time, and fluctuations in health conditions or disability symptoms are just a few of the barriers than can affect their access or interest in joining a community-based program. Wellness programs that are flexible enough to be delivered in the home setting at a convenient time of the day when participants may have lower levels of fatigue or pain present unique opportunities to reach a greater number of people.

The National Center on Health, Physical Activity and Disability (NCHPAD, https://www.nchpad.org/; accessed date: 5 September 2022), an online resource and practice Center for people with disabilities funded by the US Centers for Disease Control and Prevention since 1999, has developed three stages of wellness (Table 1) that provide people with disabilities options for how and where they would like to engage in health-enhancing behaviors. 

### 3.1. Stage I: Full Management: Health Coach-Participant Dyad

People with new and lifelong disabilities need tailored wellness programs and support services (i.e., health coaching) that address their specific physical and mental health conditions as well as social and emotional needs. For some individuals, starting with a home-based program provides them with opportunities to learn more about wellness, practice some of the techniques, and decide with their health coach which wellness domains are more important to them prior to enrolling or using a community-based wellness program (e.g., support group), venue (e.g., park), or facility (e.g., fitness center). Moreover, wellness programs that are flexible enough to be delivered in the home setting at a convenient time of the day have the potential to reach a largely underserved population of people with disabilities who may be unaware of certain areas of wellness that can improve their health or live in geographically isolated communities where access to wellness programs may be limited or nonexistent.

In Stage I, participants are offered access to a telewellness program in the comfort of their home. We recently designed a new online wellness program for adults with disabilities referred to as MENTOR—Mindfulness, Exercise and Nutrition To Optimize Resilience (https://mentor.nchpad.org/, accessed date: 12 September 2022). The program incorporates various wellness domains based on a review of the extant literature on wellness, and the customization of resources were tailored to people with disabilities.

The MENTOR program contains three primary content areas (mindfulness, exercise, nutrition) and eight secondary content areas (self-care, core values, spiritual practice, contribution to others, relationships, outdoor time in nature, rest and relaxation, and arts and leisure). The program is staffed by trained health coaches who attend a 16-h online training program that teaches them about goal setting, delivering adapted exercise, and conducting weekly health coaching classes. The MENTOR program was recently evaluated and the results found that people with disabilities who reported low wellness scores prior to the program made significant improvements in various areas of wellness including improved exercise and nutrition, better sleep patterns, and higher rates of giving back to others [31].

Stage I is described as a full-management program because the health coach provides all of the instruction to the participant. Participants are arranged in groups so that they have the benefit of socially interacting with each other through the Zoom platform. A diverse set of wellness practices are offered to participants allowing individuals to select areas of greatest need as they move across the stages of wellness. One-on-one consultations from a trained exercise, nutrition, and mindfulness instructor are available. In a recent paper, 14 people with spinal cord injury reported on what they liked about the program and they indicated that flexibility of class times, post-program support, specific exercise adaptations, and one-on-one consultations were a few notable highlights [32]. 

### 3.2. Stage II: Pivoting from Home-Based Telehealth to Community-Based Wellness

In Stage II, co-management, the focus is on a participant–provider relationship that identifies certain accommodations that the participant needs to help them consider a community-based program, or, if they are not ready to join a program, they can have additional access to an online program of interest through NCHPAD (note: a new mental health program for people with disabilities will be launched in the fall 2022). 

The focus of Stage II is to encourage and support individuals with disabilities to use community-based programs or services that will allow them to promote their health and wellness in group settings with other members with and without disability (e.g., local fitness facility, mindfulness class, diabetes management program). People with disabilities are encouraged to take a proactive approach by working with local providers to ensure that the desired program is accessible and inclusive. Participants or providers have the option of contacting a NCHPAD Expert Inclusion Specialist (EIS) who can assist them in making the program/service accessible. In cases where a participant or family member needs to find a local program, NCHPAD’s online national directory (https://www.nchpad.org/Directories, accessed date: 5 September 2022) can be used to determine if a certain program or service exists in their area. Providers may also need some type of guidance or training and also have the option to contact a NCHPAD EIS.

Since 2016, NCHPAD EIS have served as implementation facilitators in training state public health departments, program developers, professional groups, and service providers on a wide array of disability and health-promotion topics and practices through in-person and virtual opportunities. To date, the center has trained over 11,000 health and wellness professionals in such topics as:Disability awareness and inclusionInclusive worksite wellnessWalkable communities for allSupplemental nutrition assistance program and inclusionRx for exercise and healthcare providersInclusive diabetes prevention programsActive and inclusive school programs

Stage II also includes several evidence-based wellness programs that have been adapted for people with disabilities and are available to local providers interested in using one or more of these programs. A listing can be found at NCHPAD 11 Evidence-Based Adapted Programs_Final.pdf (accessed date: 5 September 2022)

### 3.3. Stage III: Wellness Self-Management

In Stage III, self-management, people with disabilities direct through their own wellness program using online resources or community-based facilities to promote their health. In the National Center on Health, Physical Activity and Disability, there are a number of web-based tools, resources, and recommendations that can be used by people with disabilities to improve a certain area of health (https://www.youtube.com/c/NchpadOrg, accessed date: 5 September 2022).

NCHPAD’s EIS are available via email, chat, or phone to address any questions related to a specific area of wellness for the individual or family member. NCHPAD also offers technical assistance service on a wide variety of topics related to physical activity, nutrition, mindfulness, disability, chronic health conditions, etc., through a 1-800 hotline, email, and live chat. To date, the Center has addressed over 15,000 requests from people with disabilities, caregivers, and health professionals. 

In 2008, NCHPAD created a self-management exercise program (14-Weeks to a Healthier You) that continues today. There have been 33,000 active users. This internet-based program was initiated as a means of providing opportunities for people with disabilities to participate in physical activity in their home with minimal or no guidance from an instructor or health coach. In the future, NCHPAD’s self-management programs will become more ‘precision-based’, allowing people to customize their own wellness program that addresses mind, body, and/or emotional/spiritual health dimensions for their specific disability (e.g., multiple sclerosis, spinal cord injury) or mobility/functional level.

## 4. Conclusions

A fragmented healthcare delivery system presents major challenges to people with disabilities and/or their family members in transitioning from healthcare services into a holistic wellness program that focuses on mind, body, and emotional/spirit-based health dimensions. 

Addressing disability inequities requires a framework that offers a continuum of wellness programs from full management health coaching to self-managed wellness directed by people with disabilities with minimal, if any, support from a health professional.

As we move forward in addressing and reducing disability disparities, we must focus on five key areas: (1) raising knowledge and awareness of the health needs of people with physical/mobility disabilities; (2) providing training and education to healthcare professionals and other providers to support a continuum of wellness services supporting adults with disabilities from home (e.g., health coaching, self-management programs) to community; (3) working closely with healthcare providers as ‘gatekeepers’ (e.g., primary care providers, physical and occupational therapists, nurses, social workers, patient navigators) who can screen and refer patients to appropriate wellness programs; (4) creating communication and dissemination resources targeted to people with disabilities and/or their caregivers and public health professionals; and (5) supporting community service providers in making their wellness programs and services inclusive for people with disabilities.

As the title of this viewpoint suggests, much of the research in disability and health over the past two decades (including my own research) has documented multiple disability disparities, and yet solutions are few and far between. While financial limitations in many countries across the world will make it challenging to adopt a similar structure as the one designed in the U.S., we can no longer spend most of our time and effort describing the disparities but doing little or nothing to manage or prevent them. People with disabilities must be able to transition from medically managed healthcare to co-managed wellness (health coach-participant dyad), to self-managed wellness (participant only), with formal transitions between each of these stages using existing resources and e-wellness programs as the essential linkages between a healthcare-to-home-to-community model.

Health professionals specializing in disability must begin to work together on a global scale to develop the infrastructure for achieving wellness equity. With online conferencing software now more ubiquitous than ever in the post-pandemic era, there are growing opportunities for professionals and people with disabilities to work together in building a system/structure that begins to address this global need. The hope is that other professionals from around the world will consider potential opportunities in their own country to share or develop similar healthcare-to-wellness models that can be described and documented so that the more than one billion people with disabilities worldwide will have a seamless pathway from healthcare to self-managed wellness.

## Figures and Tables

**Table 1 ijerph-19-11886-t001:** NCHPAD’s stages of wellness.

Stage	Stage I Full-Management	Stage II Co-Management	Stage III Self-Management
Staffing	Health Coach	Provider-Participant	Participant
Primary Content	Three core wellness domains (mindfulness, exercise, nutrition)	Risk reduction program (e.g., obesity, diabetes, hypertension, depression/anxiety) Intensive course on one wellness domain chosen by participant (e.g., exercise, meditation, cooking class)	e-Wellness content (e.g., exercise, meditation, mindfulness, cooking instruction, etc.)
Secondary Content	Eight wellness sub-domains (self-care, sleep, contribution to others, relationships, spirituality, hobbies, nature, core values)	Online consult with NCHPAD staff on community-based wellness options	
Setting	Home	Home or Community	Home
Participant Time Commitment	8 weeks @ 5 hrs./wk = 40 h	If sponsored by NCHPAD, 4–6 weeks at 2 hrs./wk = 8 to 12 h	Participant-determined

## Data Availability

Not applicable.

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
