# Peer review of "Addressing Disability Inequities: Let’s Stop Admiring the Problem and Do Something about It"

_ijerph, 2022, doi:10.3390/ijerph191911886_

Round 1
Reviewer 1 Report
The manuscript is a “commentary” and brings observations of extreme relevance to the field of knowledge. In addition, the authors share the experience of The National Center on Health, Physical Activity and Disability (NCHPAD).
Although it is clear that the approach addresses the US healthcare system, it might be interesting for the authors to present examples from other parts of the world, even though this is not a requirement for the text.
I also believe that it would be interesting to approach the concept of health present in health services, probably based on a strictly biological perspective, and which can influence the way health professionals approach.
Author Response
Comment 1. Although it is clear that the approach addresses the US healthcare system, it might be interesting for the authors to present examples from other parts of the world, even though this is not a requirement for the text.
Thank you for these comments. A statement highlighted in yellow has been added (p 4-5) that includes references to two other countries (Australia and The Netherlands) that have conducted wellness interventions for people with disabilities. After a cursory review of the literature, I was unable to find other wellness initiatives from other countries. In addition to this statement I also noted noted that there are likely other countries who have unpublished wellness initiatives/programs to ensure that the reader understands that this is only one viewpoint from one country.
I also believe that it would be interesting to approach the concept of health present in health services, probably based on a strictly biological perspective, and which can influence the way health professionals approach.
Starting at the bottom of p. 3 and p. 4 I have added a section highlighted in blue that lays out the major issues with how health is typically defined in healthcare vs. what is viewed as good (positive) health outside of healthcare.

Reviewer 2 Report
Dear Author,
The topic discussed in this article is important from the point of view of the health care system. Below are my comments on the manuscript:
1. There is no information on how many patients have already used the materials available at https://www.youtube.com/c/NchpadOrg). Has there been an evaluation - for example, has the use of online materials improved the quality of life of patients?
2. I believe that in Stages II and III there is not enough information.
3. References: Over 60% of the literature items did not come from over the last 5 years.
The article: "Addressing Disability Inequities: Let's Stop Admiring the Problem and Do Something About It" in my opinion is not a new approach to the topic, it presents relatively well-known information, and the literature is very small for this type of article and most (over 60%) not comes from over the last 5 years.
Author Response
1. There is no information on how many patients have already used the materials available at https://www.youtube.com/c/NchpadOrg). Has there been an evaluation - for example, has the use of online materials improved the quality of life of patients?
I have added some numbers regarding our current online self-management program highlighted in blue under Stage III. Under Stage II, I added the types of programs offered in the community that have been adapted by our Center and the number of professionals who have been trained. The youtube channel shows that we currently have nearly 28,000 subscribers.
2. I believe that in Stages II and III there is not enough information.
For greater clarity, I have added content to both sections highlighted in blue.
References: Over 60% of the literature items did not come from over the last 5 years.
Additional references have been added including more recent references. Unfortunately, the findings have not changed very much over the past two decades which I now state at the end of the paper and the reason why I selected the title.
The article: "Addressing Disability Inequities: Let's Stop Admiring the Problem and Do Something About It" in my opinion is not a new approach to the topic, it presents relatively well-known information, and the literature is very small for this type of article and most (over 60%) not comes from over the last 5 years.
Per my comment above, I added more contemporary references which still state the same thing over the last 20 years regarding health disparities and inequities. I have not seen any other published addressing the transition from healthcare to wellness for people with disabilities. I added a statement in the conclusion noting the emphasis of this viewpoint:
"As the title of this viewpoint suggests, much of the research in disability and health over the past two decades (including my own research) has documented multiple disability disparities and yet solutions are few and far between."

Reviewer 3 Report
This is a well written paper with many merits. The author presented a three-stage approach that has been used in a federally funded Center to address a long-standing issue (i.e., health and wellness inequities) faced by people with disabilities. Each stage was briefly and clearly described. It is encouraging to know that a telewellness program offered to the study participants at Stage 1 showed effectiveness and potential generalizability. However, it is unclear if Stages 2 and 3 were also evaluated. If so, what were the results?
There are minor suggestions for authors’ consideration:
1) The literature was relevant. However, more recent articles regarding health-related statistics should be considered (lines 37-45, p.1-2).
2) There seems to have an extra sentence in line 79 (p.2)
3) The statement “…documented in multiple studies…” in line 84 needs citations. (p.2)
4) Reference #15 is incomplete (p.6)
5) A brief description about challenges of implementing this three-stage approach is recommended.
Author Response
This is a well written paper with many merits. The author presented a three-stage approach that has been used in a federally funded Center to address a long-standing issue (i.e., health and wellness inequities) faced by people with disabilities. Each stage was briefly and clearly described. It is encouraging to know that a telewellness program offered to the study participants at Stage 1 showed effectiveness and potential generalizability. However, it is unclear if Stages 2 and 3 were also evaluated. If so, what were the results?
Thank you for this comment. In the revised paper, I added additional content and numbers highlighted in blue starting at the bottom on pages 10 and 11 related to Stages II and III.
1) The literature was relevant. However, more recent articles regarding health-related statistics should be considered (lines 37-45, p.1-2).
More recent articles have been added.
2) There seems to have an extra sentence in line 79 (p.2)
corrected
3) The statement “…documented in multiple studies…” in line 84 needs citations. (p.2)
I changed the statement to 'theoretically cut across all levels' of the socioecological model since the one citation from Martin Ginis et al. (ref 18) describes barriers at each level but only directed at PA. Some of these are based on my own experiences across a 40-year period.
4) Reference #15 is incomplete (p.6)
Corrected
5) A brief description about challenges of implementing this three-stage approach is recommended.
I have added a statement in blue (p. 13) and a concluding par. in yellow that this may be 'our time' to address these disparities at an international level using the resources in a U.S. federal center to help us work together.
"While financial limitations in many countries across the world will make it challenging to adopt a similar structure as the one designed in the U.S., we can no longer spend most of our time and effort describing the disparities but doing little or nothing to manage or prevent them."
Concluding statement (yellow):
"Health professionals specializing in disability must begin to work together on a global scale to develop the infrastructure for achieving wellness equity. With online conferencing software now more ubiquitous than ever in the post-pandemic era, there are growing opportunities for professionals and people with disabilities to work together in building an enterprise that addresses this global need."

Round 2
Reviewer 2 Report
Bez komentarza.